# A Tool for Rapid Assessment of Functional Outcomes in Patients with Head and Neck Cancer

**DOI:** 10.3390/cancers13215529

**Published:** 2021-11-03

**Authors:** Daniel Dejaco, David Riedl, Sebastian Gasser, Volker Hans Schartinger, Veronika Innerhofer, Timo Gottfried, Teresa Bernadette Steinbichler, Felix Riechelmann, Roland Moschen, Oliver Galvan, Robert Stigler, Robert Gassner, Gerhard Rumpold, Anna Lettenbichler-Haug, Herbert Riechelmann

**Affiliations:** 1Department of Otorhinolaryngology—Head and Neck Surgery, Medical University of Innsbruck, 6020 Innsbruck, Austria; daniel.dejaco@i-med.ac.at (D.D.); sebastian.gasser@student.i-med.ac.at (S.G.); volker.schartinger@i-med.ac.at (V.H.S.); veronika.innerhofer@tirol-kliniken.at (V.I.); Timo.Gottfried@tirol-kliniken.at (T.G.); teresa.steinbichler@tirol-kliniken.at (T.B.S.); praxis@hno-am-dez.at (A.L.-H.); herbert.riechelmann@tirol-kliniken.at (H.R.); 2Department of Psychiatry, Psychotherapy and Psychosomatics, Medical University of Innsbruck, 6020 Innsbruck, Austria; roland.moschen@tirol-kliniken.at (R.M.); gerhard.rumpold@tirol-kliniken.at (G.R.); 3Department of Orthopedic Surgery and Traumatology, Medical University of Innsbruck, 6020 Innsbruck, Austria; felix.riechelmann@tirol-kliniken.at; 4Department for Speech, Voice and Swallowing, Medical University of Innsbruck, 6020 Innsbruck, Austria; oliver.galvan@tirol-kliniken.at; 5Department of Oral and Maxillofacial Surgery, Medical University of Innsbruck, 6020 Innsbruck, Austria; robert.stigler@tirol-kliniken.at (R.S.); robert.gassner@tirol-kliniken.at (R.G.)

**Keywords:** head and neck neoplasms, health status, functional outcomes, surveys and questionnaires, questionnaire design

## Abstract

**Simple Summary:**

Head and neck cancer and its treatment can lead to various functional impairments. We developed and validated an instrument for rapid physician-rated assessment of basic functional outcomes in HNC patients referred to as “head and neck functional integrity scales” (HNC-FIT scales). Six basic HNC-relevant functions were identified and assigned to verbal ratings based on observable criteria. Face and content validity levels were judged adequate in systematic review by 15 experts. Validity, reliability, and responsiveness were assessed in 37 healthy controls and 84 HNC patients. All domains correlated closely with the outcome of corresponding scales of the reference questionnaire, indicating good construct and criterion validity. For all domains, interrater reliability and retest reliability were ≥0.90 and responsiveness was ≥0.15 (*p* < 0.01). Median completion time for the HNC-FIT scales was <80 s. Thus, the HNC-FIT scale appeared to be a rapid tool for physician-rated assessment of basic functional outcomes in HNC patients with good validity, reliability, and responsiveness.

**Abstract:**

Head and neck cancer (HNC) and its treatment can lead to various functional impairments. We developed and validated an instrument for rapid physician-rated assessment of basic functional outcomes in HNC patients. HNC-relevant functional domains were identified through a literature review and assigned to verbal ratings based on observable criteria. The instrument draft was subjected to systematic expert review to assess its face and content validity. Finally, the empirical validity, reliability, and responsiveness of the expert-adapted Functional Integrity in Head and Neck Cancer (HNC-FIT) scales were assessed in healthy controls and in HNC patients. A matrix of the 6 functional domains of oral food intake, respiration, speech, pain, mood, and neck and shoulder mobility was created, each with 5 verbal rating levels. Face and content validity levels of the HNC-FIT scales were judged to be adequate by 17 experts. In 37 control subjects, 24 patients with HNC before treatment, and in 60 HNC patients after treatment, the HNC-FIT ratings in the 3 groups behaved as expected and functional domains correlated closely with the outcome of corresponding scales of the EORTC-HN35-QoL questionnaire, indicating good construct and criterion validity. Interrater reliability (rICC) was ≥0.9 for all functional domains and retest reliability (rICC) was ≥0.93 for all domains except mood (rICC = 0.71). The treatment effect size (eta-square) as a measure of responsiveness was ≥0.15 (*p* < 0.01) for fall domains except for breathing and neck and shoulder mobility. The median HNC-FIT scale completion time was 1 min 17 s. The HNC-FIT scale is a rapid tool for physician-rated assessment of functional outcomes in HNC patients with good validity, reliability, and responsiveness.

## 1. Introduction

Recent advances in the diagnosis and treatment of head and neck cancer (HNC) have significantly improved the survival of affected patients and resulted in more long-term survivors [1,2,3]. Although survival is the most important aspect for HNC patients [4,5,6], the functional outcome, psychological status and health-related quality-of-life (QoL), social interactions, and economic status are becoming increasingly important as a result of this trend [7]. In a recent survey among head and neck surgeons, more than half reported that they do not systematically record functional outcomes in HNC patients. Obstacles preventing the assessment of functional outcomes included lack of support for recording and analyzing the data. The instruments were too cumbersome for many participants and the time required was too long. Many reported patient difficulties in completing questionnaires or performing tests [8].

This is contrasted by a variety of HNC-specific assessment instruments. Osborn and co-authors list 173 outcome assessment tools for HNC patients, but few are in common use [8]. The National Cancer Institute Common Terminology Criteria for Adverse Events may serve to assess adverse functional outcomes. Verbal rating scales are provided to describe severity for each adverse event term [9]. The Performance Status Scale for Head and Neck Cancer Patients is a clinician-rated instrument consisting of three subscales: normality of diet, understandability of speech, and eating in public [10]. Several HNC-specific QoL questionnaires also include functional ability subscales [8]. The revised University of Washington Head and Neck Quality-of-Life Questionnaire (UW-QoL) is a common QoL instrument in the US. The major parts consist of functional assessment scales [11,12,13]. The Functional Assessment of Cancer Therapy—Head and Neck Scale [14] is primarily about well-being and not about assessing functional deficits; however, HNC-specific questions also ask about functional aspects. The European Organization for Research and Treatment of Cancer Head and Neck Questionnaire 35 (EORTC QLQ-H&N35) serves as an HNC-specific QoL questionnaire. It is used together with the basic EORTC QLQ-C30. The EORTC QLQ-H&N35 contains 30 questions related to functional outcomes in HNC patients [15]. The most extensive functional assessment tool is the International Classification of Functioning, Disability, and Health (ICF). It provides a description of situations with regard to human functioning and its restrictions and serves as a framework to organize this information [16]. Due to the holistic approach and the high complexity, the ICF is not suitable for routine clinical recording of functional outcome at oncological follow-up visits. Although core sets have been extracted for clinical application in HNC patients [17], to our knowledge there is currently no clinically applicable and formally validated instrument for the assessment of functional outcomes based on the ICF.

The aim of this study was to develop and validate a rapid clinical tool to assess basic physical and mental functional outcomes of HNC patients during routine oncology follow-up. The tool should be limited to a few relevant HNC-related functional domains and avoid redundancy. Scores should be based on external, observable indicators that can be ranked, with ranks ideally equally distributed across functional domains. Rating scales should be positively aligned so that low scores represent functional limitation, with high scores representing functional integrity.

## 2. Materials and Methods

The head and neck functional integrity (HNC-FIT) scales was developed in a stepwise approach, as recommended by the Quality-of-Life Group of the EORTC [18]. Since these guidelines were created for the development of patient-reported outcomes, phases were adapted accordingly. The instrument was developed in 4 phases [18,19]. In the first phase, functions and symptoms most relevant for HNC patients were identified and condensed into a few functional domains. In the second phase, observable criteria for a uniform scoring system were developed and a draft of the HNC-FIT scales was created. In the third phase, draft HNC-FIT scales were evaluated and adapted by a group of HNC experts from different disciplines and professions. Finally, the adapted HNC-FIT scales were formally validated in healthy controls and HNC patients. The study was approved by the ethics committee of the Medical University of Innsbruck (1182/2019).

### 2.1. Identification of HNC-Relevant Functional Domains

A literature search was performed using the search terms “head and neck cancer”, “functional outcome”, and “questionnaire” in the National Library of Medicine database. All English-language studies between 1985 and 2020 were reviewed. In addition, commonly used outcome instruments were evaluated. Study titles and abstract were screened for relevance. Studies considered not relevant to the topic were excluded. Exclusion criteria were (a) irrelevant primary tumor site (e.g., esophagus, colorectal, thyroid, parathyroid, skin, lung, bladder, soft-tissue, brain), (b) irrelevant histology (e.g., lymphoma, mucosal melanoma, mesothelioma, retinoblastoma), or (c) other reasons (e.g., pediatric patients, record did not explore cancer, record not in English). Functions and symptoms that occurred in the title, abstract, or full text of multiple publications were identified and assigned to as few higher-level HNC-specific functional domains as possible, such as oral food intake (eating and drinking), breathing, or speech (voice and articulation). Functional domains were chosen to avoid overlap with other functional domains, i.e., to have low redundancy.

### 2.2. External Criteria for Uniform Scoring and Draft Development

Functional integrity was recorded ordinally on verbal rating scales. Verbal ratings ranged from phrases implying complete loss of function or worst functional outcome to normal function, i.e., functional integrity. Normal function was defined as the individual functional status before the onset of disease. As with the UW-QoL and CTCAE approaches, verbal ratings should not reflect the patients’ or physicians’ subjective impression but should be anchored to external observable criteria to provide some degree of objectivity [9,13]. The wording was chosen so that the levels reflected the extent of functional impairment as evenly as possible across all functional domains (equidistance). Numerical scores were assigned to the verbal ratings, ranging from zero for the worst outcome to the highest number for functional integrity (positive scale). To allow rapid completion by ticking by the examining clinician, the functional domains and their verbal ratings were arranged in a matrix. Finally, a draft and instructions on how to complete the HNC-FIT scales were created.

### 2.3. Draft Revision through Semi-Structured Expert Interviews

Face and content validity were assessed by structured interviews with experts in the diagnosis and treatment of HNC patients or experts in one of the identified functional domains. Good face validity was assumed if the mean of the expert rating for global plausibility was less than 2 (good) on a Likert scale of 1 (very good) to 5 (not sufficient). For content validity, each functional domain was assessed with 8 questions on the same 5-point Likert scale. For each of these questions, the scores were recorded, along with the experts’ comments and suggestions for improvement. If the average of the experts mean score of the 8 questions for a functional domain was above 2, the functional domain was discussed in detail with the experts and corrections were made as necessary. Corrections had to be consistent with the main intention of the instrument to capture functional outcome, not well-being. Requested additional functional domains had to occur with sufficient frequency in the performed literature search and had to be recordable in verbal ratings using observable criteria. In a complex coordination process, the adapted version of the HNC-FIT scales was then created.

### 2.4. Empirical Validation of the Adapted HNC-FIT Scale

The adapted HNC-FIT scale was a singly ordered matrix of six functional domains with five levels each. The clinician used this matrix as a template for a structured patient interview and ticked the appropriate functional scores in the matrix.

#### 2.4.1. Patients and Controls

HNC patients from the University Department of Otorhinolaryngology—Head and Neck Surgery, Medical University of Innsbruck, Austria, were asked to participate in the validation of the HNC-FIT scales. Inclusion criteria for HNC patients were age ≥18 years and histologically confirmed HNC from the oral cavity, oropharynx, larynx, hypopharynx, or carcinoma of unknown primary with any UICC stage. Patients with cognitive impairment were excluded. Patients with incident HNC before treatment were prospectively recruited in the order of their arrival at the department’s outpatient clinic in 2018 and 2019 (pretreatment group). Patients with incident HNC after curative treatment were recruited in the same way during oncology follow-up (posttreatment group). Various clinical characteristics such as age; sex; histology; tumor location; UICC stage; and T-, N-, and M- stages were recorded. Feeding tubes were always placed via percutaneous endoscopic gastrostomy.

Control subjects were approached in the cafeteria of the University Hospital Innsbruck in approximately the known age and sex distribution of HNC patients and asked if they were currently healthy and if they would be willing to participate as controls in this study. The cafeteria is frequented by academic staff, nursing staff, administrative staff, and workers. The age and gender of controls were recorded. Informed consent was obtained from all volunteers.

#### 2.4.2. Empirical Validity, Reliability, Responsiveness, and Fill-In Time

Empirical validity was assessed in two ways. For construct validity, it was tested if the HNC-FIT scales behave as expected. Best outcomes were expected in healthy controls, followed by patients with incident HNC before treatment, followed by HNC patients after treatment. To assess this trend, mean ranks of the three participant groups for each functional domain were calculated and tested using the Jonckheere–Terpstra trend test with correction for ties. To assess criterion validity, the EORTC QoL H&N35 was completed by the HNC patients. The German version of the EORTC H&N35 was provided by the EORTC according to a license agreement. Spearman’s correlation coefficients were calculated between each functional domain and corresponding EORTC H&N35 subscales and items. While for the HNC-FIT scales for food intake, speech, and pain, clearly corresponding EORTC H&N35 subscales were available, no corresponding subscales for breathing, mood, or neck and shoulder mobility were available.

Two types of reliability were assessed. To assess interrater reliability, two physicians consecutively completed the adapted HNC-FIT scales in random order, blinded to each other. To assess retest reliability, the HNC-FIT scales were reassessed 5 to 10 days after the last assessment in posttreatment patients by the same rater. For interrater and test–retest reliability in posttreatment patients, intraclass correlation coefficients (ICC) were calculated using a two-way mixed effects, absolute agreement, same raters model.

Responsiveness was assessed in pretreatment patients, since in these patients treatment-related changes in functional integrity were to be expected. HNC-FIT scales were completed before treatment and again at the first follow-up visit after the end of treatment (i.e., after surgery only or after multimodality treatment consisting either of primary radiochemotherapy or of surgery followed by postoperative radiation). For responsiveness, we calculated a repeated measures ANOVA before and after treatment. Partial eta square as a measure of effects size served to evaluate responsiveness, with values of η2 = 0.01, η2 = 0.06, and η2 = 0.14 indicating small, medium, and large effects, respectively [20]. Finally, the time to complete the HNC-FIT scales was recorded with a stopwatch.

#### 2.4.3. Sample Size Estimation

The sample size estimate for assessment of responsiveness was based on a repeated-measures ANOVA with two time-points. The α-error was set to 0.05, the β-error to 0.2, r to 0.5, and f to 0.25, resulting in a sample size of 34 HNC patients for the responsiveness study. For assessment of validity, the sample size for mean differences was calculated with a large effect size with three groups (HNC patients before treatment, after treatment, and controls). An ANOVA with identical parameters as above was assumed for this analysis. This resulted in a total sample size of 64 subjects. Sample size estimates were calculated using GPower3.1 [21]. For sample size estimates of intraclass correlation coefficients, the online sample size calculator provided by Arifin was used [22], assuming an expected reliability of 0.8 ± 0.15, an α-error of 0.05, and a power of 0.8, resulting in a sample size of 24.

### 2.5. Data Analysis

Frequencies, nominal data, and ordinal data were tabulated. For interval scaled data, means and standard deviations were calculated if not stated otherwise. The scores for each functional domain were dichotomized into normal and near-normal functional outcomes (numerical scores 3 and 4) vs. impaired functional outcomes (numerical scores 0–2). The percentage of patients achieving normal or near-normal functional outcomes (functional integrity) were calculated and depicted in a star plot. For multiple comparisons, *p*-values were corrected using the Holms–Bonferroni method [23] to avoid risk of type I error. Statistical analyses were performed using SPSS 27 (IBM, Armonk, NY, USA) if not stated otherwise.

## 3. Results

### 3.1. HNC-Relevant Functional Domains

The literature search yielded 1273 records. Of these, 120 articles complied with exclusion criteria and were subjected to full text analysis (Appendix A), together with the previously identified publications. The full text analysis revealed 39 functions and symptoms, which were initially assigned to 7 functional domains. These were oral food intake, saliva, respiration, speech, pain, mood, and neck–shoulder mobility (Appendix A). As in the International Classification of Functioning, Disability, and Health, pain and mood were conceived as functions [16].

### 3.2. External Criteria for Uniform Scoring

Observable, external criteria for creating an ordered set of verbal ratings included dependence on a feeding tube and normality of diet for the oral food intake domain, dependence on a tracheotomy and dyspnea for the breathing domain, pain medication for the pain domain, need for antidepressants and feeling depressed for the mood domain, and problems combing hair and looking backward while driving for the neck and shoulder mobility domain. The wording of the verbal rating scales implicating a reasonably uniform ordering of functional integrity allowed five levels of functional integrity for each functional domain (Appendix A). Despite considerable efforts, we were not able to code the functional domain for dry mouth, chewing, and dental status using external criteria with verbal rating scales in a meaningful sequence (Appendix A). This functional domain was assigned to the functional domain food intake (Appendix A). Finally, a draft with a matrix of six functional domains with five levels each was created for expert review (Appendix A). Detailed instructions on how to interpret items and how to complete this preliminary version of the HNC-FIT scale were provided on the back of the form (Appendix A). HNC-related symptoms and functions, which could not be coded to functional domains or could not be operationalized in a meaningful way, are presented as Appendix A.

### 3.3. Draft Revision through Semi-Structured Expert Interviews

The draft HNC-FIT scale was reviewed by 17 experts in the multidisciplinary treatment of HNC patients (Appendix A). Of these, 9 were women. The average length of professional experience in each discipline was 15.7 ± 5.5 years. The expert team consisted of 2 otolaryngologists, 2 maxillofacial surgeons, 2 medical oncologists, 1 radiation therapist, 1 orthopedist, 2 phoniatrics, 1 psychologist, 1 anesthesiologist, 1 physical therapist, and 2 speech therapists. The mean score for global acceptance (1.7 ± 0.6) met the predefined threshold suggesting acceptable face validity. With regard to content validity, experts pointed out that several functional impairments assessed with the HNC-FIT scale can also be caused by conditions other than HNC, e.g., preexisting depression or other causes of chronic pain. Therefore, the remark “due to tumor/treatment” was added to HNC-FIT scales (Appendix A). Additional domains were suggested by some experts, including social interaction; QoL; sleep quality; aesthetic appearance; and dry mouth, chewing, and dental status domains. In intense discussions, it was found that the suggested additional domains either did not meet the main intention of capturing functional outcomes instead of well-being, did not occur frequently enough in the literature review, or could not be operationalized by verbal ratings anchored to external criteria. Based on expert advice, the wording of verbal rating scales was changed for food intake, speech, pain, mood, and shoulder–neck mobility.

### 3.4. Empirical Validation of the Expert Adapted HNC-FIT Scale

#### 3.4.1. Patients and Controls

A total of 37 volunteers who considered themselves healthy (controls), 24 pretreatment HNC patients, and 60 posttreatment HNC patients kindly agreed to participate in the evaluation the HNC-FIT scale.

Of the 24 pretreatment HNC patients, 13 received surgical treatment only and 11 received multimodality treatment. For all pretreatment HNC patients, the mean interval between first and second assessment was 55 (±35; range 15–136) days, if receiving surgery only 29 (±10; range 15–50) days, and if receiving multimodality treatment 85 (±28 days, range 34–136) days.

Of the 60 posttreatment HNC patients, 8 received no surgery only, 15 received surgery only, and 37 received multimodality treatment. Of these, approximately half of the patients (*n* = 32) received their treatment within the last two years and the other half (*n* = 28) within the last 5 years since inclusion. Since all posttreatment HNC patients were assessed and re-assessed within 5 to 10 days, the mean interval was 9 (±4; range 3–18) days. Patient and disease characteristics are summarized in Table 1.

#### 3.4.2. Empirical Validity, Reliability, Responsiveness, and Fill-In Time

As expected, average ranks of HNC-FIT-scores descended from controls to pretreatment to posttreatment patients (Table 2), supporting the construct validity of the HNC-FIT scales.

In the pain and mood domains, there was a significant step from controls to pre- and posttreatment patients (*p* < 0.01). Similarly, a significant step from control and pretreatment patients to posttreatment average ranks (*p* = 0.027) was observed in the neck and shoulder mobility domain. HNC-FIT scales correlated with the corresponding EORTC QoL H&N35 subscales supporting criterion validity. The HNC-FIT scale food intake had the highest correlations with the H&N35 subscales ”feeding tube” (r = −0.73, *p* < 0.001), ”swallowing” (r = −0.72, *p* < 0.001), and ”social eating” (r = −0.56, *p* < 0.001). For the HNC-FIT scale “speech”, the highest correlations were found with the H&N35 subscale ”speech” (r = −0.55, *p* < 0.001), and for the HNC-FIT scale “pain” with the H&N35 subscales ”pain” (r = −0.47, *p* < 0.001) and ”pain killers” (r = −0.61, *p* < 0.001; Table 3).

Intraclass correlation coefficients (rICC) for interrater reliability ranged from 0.90 to 0.99 (Table 4). Intraclass correlation coefficients for evaluation of the test–retest reliability between two measurements in posttreatment HNC patients were well above 0.9 for all functional domains except mood, which had an rICC of 0.71 (Table 4).

To evaluate the responsiveness of the adapted HNC-FIT scale, mean changes before and after treatment for pretreatment HNC patients were compared. The partial eta square, a measure of the effect size in a repeated measures ANOVA before and after treatment, served to evaluate responsiveness. The partial eta square values suggested good responsiveness for food intake (η2 = 0.31; *p* = 0.040) and pain (η2 = 0.56; *p* = 0.006). However, no significant responsiveness was observed for breathing (η2 = 0.10; *p* = 0.33), speech (η2 = 0.26; *p* = 0.08), neck and shoulder mobility (η2 = 0.01; *p* > 0.99), or mood (η2 = 0.14; *p* = 0.27; Table 5). The median completion time was 1 min and 17 s (25th percentile: 54 s; 75th percentile: 1 min and 47 s). The shortest completion time was 16 s and the longest was 3 min and 26 s.

## 4. Discussion

### 4.1. Introduction

Measures of HNC outcomes include survival, HNC-related functional integrity, health-related QoL, and economic and social statuses. In a recent survey among head and neck surgeons, more than half reported that they do not systematically record functional outcomes, in part because the instruments are too cumbersome and the time required is too long. In contrast to other common otorhinolaryngologic diseases such as chronic rhinosinusitis [25,26], simple, clinically applicable instruments for recording HNC-related basic physical and mental functions are rare [8,27]. The aim of this study was to develop and validate an instrument to assess a basic functional status of HNC patients at oncological follow-up visits. The tool should be able to be applied by the clinician with the least amount of time possible. Only higher-level physiological and mental functional domains such as sight, smell, taste, hearing, pain, mood, food intake, breathing, or speech should be recorded (Appendix A). Detailed functions and symptoms related to these functional domains were intentionally omitted (Appendix A). In addition, only functional domains that are commonly affected in HNC patients should be recorded and redundancies should be avoided.

### 4.2. Development

The HNC-FIT scales were developed in 4 phases [18,19]. To identify HNC-relevant functional domains, we conducted a literature search on functional outcomes in HNC and initially assigned the most frequently recurring keywords to 7 functional domains. These domains were food intake, saliva, breathing, speech, pain, mood, and shoulder–neck mobility. Other HNC-related functional domains such as vision, hearing, smell, sleep, fatigue, appetite, body image, sexuality, cognitive functioning, anxiety and worry, and social and occupational status did not occur frequently enough in the analyzed publications to be considered. To maintain clarity of the HNC-FIT scales, these domains were not included and should be reserved for more comprehensive functional assessment tools.

The next phase was for operationalization according to the objectives of the study. The initially identified 7 functional domains were arranged in a matrix of similarly graded verbal rating scales from loss of function to normal function. It was found that 5 functional levels could be plausibly formulated into verbal ratings that were linkable to external criteria and had comparable spacing between levels and across functional domains. The dry mouth, chewing, and dental status function could not be operationalized in this way. For “dry mouth”, an attempt was made to use the frequency of oral fluid intake due to dry mouth as an external criterion in a pilot study. However, this parameter depended on numerous factors unrelated to saliva production. In addition, this parameter was partially redundant with the functional domain food intake and was omitted. This resulted in a draft of 6 verbal rating scales with 5 levels each, for which detailed instructions for clinicians on how to complete it were established. The verbal rating scales were also numerically coded in the sense of a positive 5-point Likert scale from 0 for loss of function or worst outcome to 4 for normal function.

In the third phase of instrument development, this draft was evaluated by 17 experts for face and content validity. Experts were either involved in the multidisciplinary treatment of HNC patients or experts for specific functional domains. While the whole concept of HNC-FIT scales was generally accepted, the experts recommended clarifying whether preexisting, tumor-independent functional limitations should be considered or only functional limitations caused by the HNC. Therefore, the information “due to tumor/treatment” was added to the right of the scales (Appendix A). Additional functional domains, particularly social function, were suggested by several experts, although corresponding keywords occurred comparatively infrequently in the literature search. Moreover, revision of the functional domains food intake, speech, mood, and shoulder–neck mobility was suggested. Proposed changes were discussed in detail with the experts and adapted to meet the aims of capturing higher-level functional domains rather than specific functions, capturing functional outcomes rather than QoL, and formulating uniformly graded verbal ratings based on external criteria. Several expert suggestions were also included in the instructions for clinicians on how to complete the Head and Neck Functional Integrity Scale (Appendix A).

### 4.3. Mode of Administration

The mode of administration of outcome assessment instruments may have an impact on the results and data quality [28,29,30]. We decided against a patient-reported assessment and in favor of a clinician-based assessment. First and foremost, we had concerns about whether patients would be able to correctly understand the verbal ratings. Even physicians needed some explanation and training on how to fill in the HNC-FIT scales. In addition, physician-based assessments have several advantages. The examiner can base their assessment on the medical history, patient interview, and physical examination. In addition, the attending physician can inquire whether anything is unclear or specifically obtain their own assessment of a functional limitation. The clinician-based assessment also ensures that the examiner is aware of the patient’s functional limitations and can take appropriate rehabilitative measures. Finally, a certain plausibility check takes place and language barriers or problems with writing or reading can be better resolved.

### 4.4. Empirical Validation

For empirical validation, the adapted HNC-FIT scale (Appendix A) was evaluated in 37 controls, 24 patients with incident HNC before treatment, and 60 posttreatment HNC patients in oncology follow-up. Controls had approximately the age and sex distribution of HNC patients and considered themselves healthy. It is likely that hospital employees are not representative of the general population of the same age and sex, although no relevant bias was suspected here with respect to the 6 functional domains studied. The included HNC patients were recruited in order of entrance at the clinic and were reasonably representative for HNC patients in our region (Table 1).

To assess the psychometric properties of the adapted HNC-FIT scales, standard techniques were employed [19,25]. As expected, there was a descending trend of functional outcomes from controls to pretreatment to posttreatment HNC patients (Table 2). Considering the ordinal level of HNC-FIT scales, mean ranks and the Jonckheere–Terpstra test seemed appropriate for this question. The results of this known-group comparison supported the construct validity of the HNC-FIT scales. The expected trend across the 3 participant groups was highly significant for all domains except mood and pain. However, in the mood domain, controls scored highest (*p* < 0.01) and in the pain domain, posttreatment patients scored lowest (*p* < 0.01; Figure 1).

It is also reasonable that incident HNC patients do not have a better mood before therapy, i.e., directly after diagnosis, than following therapy. Comparison with the functional scales of the EORTC QoL H&N35 served to assess criterion validity. Good psychometric properties have been reported for this disease specific QoL questionnaire, which is frequently used in Europe (30). All HNC-FIT scales correlated well with the corresponding patient-reported subscales of the EORTC QoL H&N35 (Table 3). The reliability of the HNC-FIT scales was estimated with intraclass correlation coefficients (rICC) [19]. Overall, excellent interrater reliability was observed for all functional domains, with rICC scores above 0.9 for all functional domains (Table 4). In addition, excellent values were also observed for retest–reliability for all functional domains [24], except for the functional domain mood, which was in an acceptable range with an rICC of 0.71 (Table 4). Internal consistency (Cronbach’s alpha) was not used as a measure of reliability because it implies some redundancy of items [31], which was intentionally avoided. Good responsiveness was observed for the functional domains of food intake, speech, pain, and mood (all *p* < 0.05) (Table 5). Interestingly, this was not observed for respiration (*p* = 0.17) or shoulder–neck mobility (*p* = 0.67). Most patients available for the responsiveness tests had received primary radio- or chemoradiotherapy. Impaired neck and shoulder mobility after radiotherapy often develops within months. This was outside the observation period used to assess responsiveness in this study. In addition, respiratory problems or the need for a tracheostomy are less likely to occur in these patients. Finally, a median completion time of 1 min and 17 s for the HNC-FIT scales is considered acceptable, even given the lack of time during oncology follow-up. The 6 score values can be easily entered directly into the clinical information system.

### 4.5. Presentation of Functional Outcomes

A direct and comprehensive presentation of results is the plain listing of the absolute frequencies with which each verbal rating was marked (Table 6).

Relative frequencies are co-determined by the number of participants in each group. Relative frequencies of verbal ratings in percent for the numbers of participants per group can be plotted in a stacked bar chart (Figure 2).

Figure 2 reveals a fairly even distribution of scores across the functional domains within each participant group. This suggests that the formulation of the verbal ratings achieved the goal of uniform scaling across functional domains reasonably well. On the other hand, significant differences are evident between participant groups, which supports the good construct validity. To further ease the outcome interpretation, scores were dichotomized in scores of 0 to 2 (impaired function) vs. 3 and 4 (functional integrity). The outcomes in the control group, where scores <3 occurred in only 2.5%, supported this cut-off (Figure 2, left panel). This dichotomization allowed a concise presentation of the basic HNC-relevant functional outcomes in a star graph (Figure 1). It can be easily recognized that for the mood domain, functional integrity was found in only 92% of controls, whereas it was found in close to 100% of controls in all other functional domains. Mood, pain, and food intake were the functional domains most frequently impaired in HNC patients before treatment, whereas the other domains were only rarely affected. In posttreatment HNC patients, functional integrity was observed in 80–90% of patients for mood, pain, and neck and shoulder mobility. However, in the speech, breathing, and food intake domains, functional integrity was considerably less frequent. Normal or near-normal food intake was achieved in less than 70% of posttreatment HNC patients. It is, however, important to consider that these posttreatment results were obtained in a small group of unselected HNC patients for all tumor sites, stages, and treatment modalities. They serve only to demonstrate possible modes of outcome presentation. The sample size was by far too low to draw any general conclusions on functional outcomes in HNC. Mean scores for the 6 functional domains of the 3 study groups are listed in Appendix A; however, the mean scores depend on the scaling of each specific outcome assessment instrument, are difficult to compare across different instruments, and are considered less intuitive than the percentage of patients achieving normal or near-normal functional outcomes.

### 4.6. Limitations

The main advantage of the HNC-FIT scales as compact, rapid instruments is also their main limitation. By restricting the scales to the functions and symptoms most frequently mentioned in publications, many important functional domains such as hearing or balance are ignored. This also applies to the assessment of QoL, which was suggested by experts involved in the semi-structured interviews. In intense discussions, it was found that the suggested QoL domain did not meet the main intention of capturing functional outcomes. In addition, it was suggested to include the assessment of dry mouth, chewing, and dental status. Although all three suggestions were considered relevant, despite a considerable effort no meaningful operationalization could be performed (i.e., anchoring to external criteria and equidistance between verbal ratings) (Appendix A). When functions and symptoms are combined into functional domains for the sake of simplicity, detailed information is certainly lost. This can include dry mouth, chewing, and dental status information, which was subsumed under the higher-level functional domain of oral food intake. The number of functional items and the level of detail are a compromise between the desired characteristics of the outcome assessment instrument and clinical applicability. However, various single function assessment tools are available and may supplement the HNC-FIT scales if required (e.g., visual analogic scale to assess QoL, short QoL screeners such as the EQ-5D [32] or the chewing function questionnaire [33]).

As with other outcome assessment instruments, the HNC-FIT scales are subject to various forms of bias [19]. Anchoring assessments to observable external criteria reduces susceptibility to bias. Thus, it can be determined largely without bias whether or not a patient has a tracheostoma or a feeding tube. However, for some items, the investigator relies on the patient’s information, which may lead to bias, e.g., due to social desirability. Likely a more relevant source of bias is the halo phenomenon, i.e., that the examiner’s overall impression of the patient influences the assessment of individual functional domains. Probably the most important cause of bias in the HNC-FIT scales is that filling them out is cumbersome for both the clinician and the patient and should be done as quickly as possible. The fastest way is to check “normal” for all functional domains. This saves the time needed to check the extent of impairment in detail.

Some additional limitations of the present study ought to be discussed. Firstly, the standardization of the temporal collection of the data was suboptimal. For pretreatment patients, fixed time-points were defined during conceptualization of the study (before treatment and again at the first follow-up visit after end of treatment). However, these intervals significantly varied between patients receiving surgery only (approximately 4 weeks) and patients receiving multimodality treatment (approximately 14 weeks). The latter treatment requires significantly more pretreatment work-up (at our institution usually 4 weeks [34]) and time (usually 6 to 8 weeks) to complete treatment. All pretreatment patients were prospectively recruited in the order of their arrival at the outpatient department. Both the treatment recommendation and consequently the time-point of the first follow-up visit after the end of treatment were based on the recommendation of the institutional interdisciplinary tumor board. Thus, no further optimization of temporal data collection could be achieved. For posttreatment patients, the interval between the first and second assessment was relatively constant at 5 to 10 days. However, the time of assessment during oncologic follow up was not fixed. Therefore, half of the patients were assessed within the first two years and the other half within the third to fifth years after end of treatment. It is probable that functional integrity scores raised with the HNC-FIT scales are influenced by the time-point of assessment during oncologic follow-up. Unfortunately, the number of patients included in this study was too small to observe a significant difference within the posttreatment group (*p* > 0.12).

Secondly, it is also likely that the stage, type of therapy and tumor site influence functional integrity scores raised with the HNC-FIT scales. Unfortunately, the number of patients included in this study was too small to conduct multivariable analyses to investigate the unique contribution of each variable on the total score or domain scores.

Thirdly, the HNC-FIT scales were developed and empirically validated for German-speaking patients only. Although the original German versions of the HNC-FIT scales were translated to English by the authors themselves (Appendix A), the translated HNC-FIT scales had neither been professionally translated nor empirically validated for English-speaking patients. The original German versions of the HNC-FIT scales are provided as Appendix A (Appendix A).

## 5. Conclusions

The HNC-FIT scale is a plain tool for rapid assessment functional outcomes in HNC patients with good psychometric properties. It allows for quick capture and clear presentation of key functional results, filling a gap in HNC outcome assessment.

## Figures and Tables

**Figure 1 cancers-13-05529-f001:**
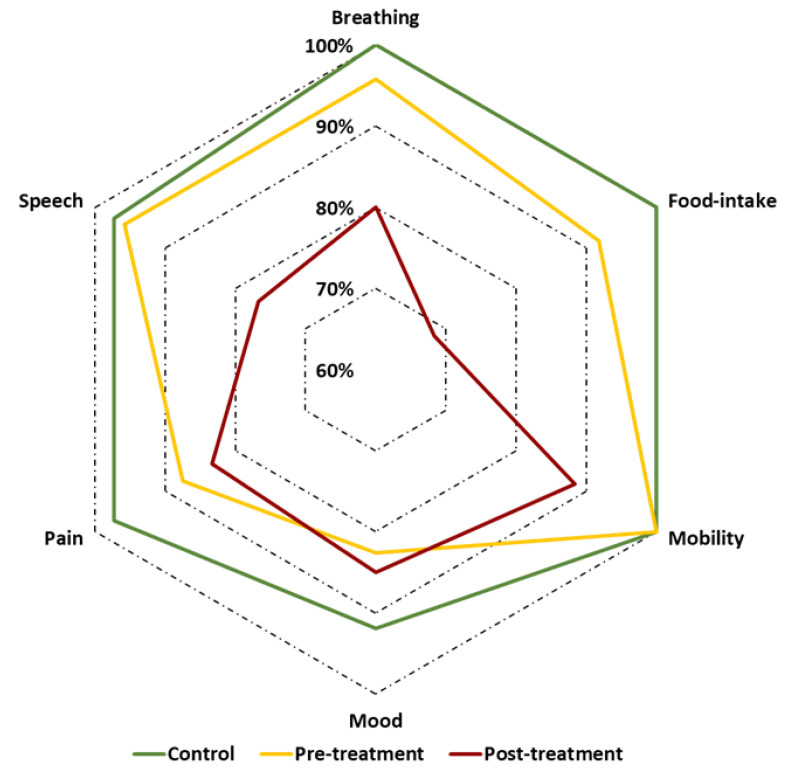
Star plot showing the percentages of normal and near-normal scores (functional integrity; scores 3 and 4) for controls (green line), HNC patients before therapy (yellow line), and HNC patients after therapy (red line). The star axes represent the percentages of study participants with functional integrity for the functional domains breathing, food intake, neck and shoulder mobility, mood, pain, and speech.

**Figure 2 cancers-13-05529-f002:**
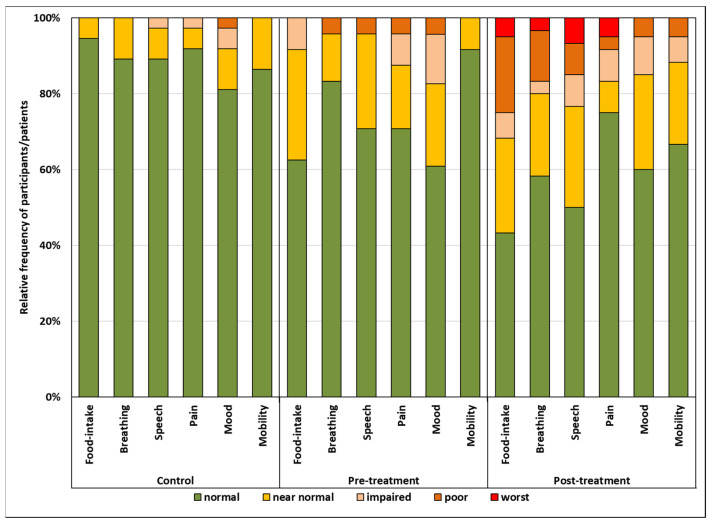
Relative frequencies (percent) of integrity scores in 6 functional domains in controls, HNC patients before treatment, and HNC patients after treatment. Mobility refers to neck and shoulder mobility.

**Table 1 cancers-13-05529-t001:** Clinical data for HNC patients and controls.

	Controls (*n* = 37)	Pretreatment Group (*n* = 24)	Posttreatment Group (*n* = 60)
**Sex**			
Male	25	21	49
Female	12	3	11
**Age**			
≤50	4	3	3
51–60	13	7	12
61–70	16	7	19
71–80	4	4	19
≥80	0	3	5
**P16 status**			
Negative	n.a. ^4^	20	49
Positive	n.a. ^4^	4	11
**Tumor site**			
Oral cavity	n.a. ^4^	5	12
Oropharynx	n.a. ^4^	6	25
Hypopharynx	n.a. ^4^	4	8
Larynx	n.a. ^4^	8	13
CUP ^1^	n.a. ^4^	1	2
**Clinical UICC stage ^2^**			
Stage I	n.a. ^4^	10	13
Stage II	n.a. ^4^	5	14
Stage III	n.a. ^4^	2	9
Stage IVa	n.a. ^4^	6	21
Stage IVb	n.a. ^4^	1	3
Stage IVc	n.a. ^4^	0	0
**Treatment ^3^**			
No surgery only	n.a. ^4^	0	8
Surgery only	n.a. ^4^	13	15
Multimodality	n.a. ^4^	11	37

^1^ CUP: cancer of unknown primary; ^2^ UICC: Union for International Cancer Control; ^3^ Treatment: no surgery only included primary radiation or primary concurrent radiochemotherapy without any surgical intervention; surgery only includes surgical treatment of HNC without any postoperative radiation or radiochemotherapy, multimodality treatment includes surgery follow by postoperative radiation or radiochemotherapy followed by salvage surgery; ^4^ n.a.: not applicable.

**Table 2 cancers-13-05529-t002:** Mean ranks in the Jonckheere–Terpstra tests for 6 functional domains in controls, HNC patients before treatment, and HNC patients after treatment.

Functional Domain	Controls	Pretreatment Group	Posttreatment Group	*p*-Value ^1^
Food intake	80.8	63.5	47.9	<0.001
Breathing	71.7	67.8	51.7	0.004
Speech	74.4	65.3	50.4	0.001
Pain	68.9	56.5	57.9	0.140
Mood	69.3	56.8	56.7	0.140
Mobility	66.8	69.8	53.9	0.027

^1^ Holm–Bonferroni-corrected *p*-values.

**Table 3 cancers-13-05529-t003:** Spearman correlation coefficients of the clinician-rated HNC-FIT scale and the 18 corresponding patient-rated EORTC QoL H&N35 subscales in 84 patients with HNC.

		HNC-FIT Scale					
		Food Intake	Breathing	Speech	Pain	Mood	Mobility
**EORTC QoL H&N35**	Teeth	−0.16	−0.01	−0.17	−0.13	**−0.32 ^3^**	−0.09
	Swallowing	**−0.72 ^1^**	−0.20	−0.31 ^3^	−0.31 ^3^	−0.22	**−0.41 ^2^**
	Social eating	**−0.56 ^1^**	0.02	−0.20	−0.24	−0.15	**−0.42 ^2^**
	Opening mouth	−0.36 ^2^	−0.05	−0.21	−0.24	−0.26	**−0.40 ^2^**
	Dry mouth	−0.29 ^3^	0.15	0.10	−0.12	−0.06	−0.21
	Sticky saliva	−0.38 ^2^	−0.13	−0.12	−0.11	−0.26	−0.32 ^3^
	Nutritional Supplements	−0.35 ^2^	−0.02	−0.09	−0.15	−0.09	−0.07
	Feeding tube	**−0.73 ^1^**	−0.20	−0.32 ^3^	−0.15	0.10	−0.26
	Weight loss	−0.38 ^2^	−0.15	−0.21	−0.26	−0.01	−0.02
	Weight gain	0.04	−0.26	−0.26	0.05	−0.16	−0.23
	Coughed	−0.15	−0.16	−0.15	0.03	−0.11	0.05
	Speech	−0.34 ^2^	**−0.37 ^2^**	**−0.55 ^1^**	−0.35 ^2^	−0.28	−0.11
	Pain	−0.36 ^2^	0.04	−0.16	**−0.47 ^1^**	−0.24	−0.18
	Pain killers	−0.35 ^2^	−0.05	−0.16	**−0.61 ^1^**	−0.21	−0.05
	Social contact	−0.01	−0.03	−0.09	−0.29 ^3^	−0.25	−0.08
	Sexuality	−0.27 ^3^	**−0.39 ^2^**	−0.30 ^3^	−0.05	−0.25	−0.22
	Senses	−0.37 ^2^	−0.19	−0.25	−0.19	−0.19	−0.26
	Feeling ill	−0.26 ^3^	−0.14	−0.09	−0.11	**−0.30 ^3^**	−0.21

^1^*p* < 0.001; ^2^
*p* < 0.01; ^3^
*p* < 0.05; highest correlations displayed in bold. The negative signs are caused by the opposite scale arrangements. Mobility stands for neck and shoulder mobility. Holm–Bonferroni-corrected *p*-values.

**Table 4 cancers-13-05529-t004:** Reliability of the adapted HNC-FIT scale.

		rICC ^1^	95% CI ^2^	*p*-Value ^2^
**Interrater Reliability**	Food intake	0.99	0.98–0.99	0.006
	Breathing	0.95	0.93–0.97	0.006
	Speech	0.96	0.94–0.98	0.006
	Pain	0.93	0.89–0.95	0.006
	Mood	0.91	0.86–0.94	0.006
	Shoulder–neck mobility	0.90	0.85–0.93	0.006
**Test-Retest-Reliability**	Food intake	0.94	0.84–0.98	0.006
	Breathing	0.95	0.91–0.99	0.006
	Speech	0.97	0.94–0.98	0.006
	Pain	0.93	0.81–0.98	0.006
	Mood	0.71	0.18–0.91	0.012
	Dhoulder–neck mobility	0.98	0.93–0.99	0.006

Note: ^1^ rICC = Intraclass correlation coefficient; ^2^ CI = confidence interval; rICC > 0.7 = acceptable, > 0.8 = good, > 0.9 = excellent [24]. ^2^ Holm–Bonferroni-corrected *p*-values.

**Table 5 cancers-13-05529-t005:** Responsiveness values of the HNC-FIT scale.

Functional Domain	Pretreatment Assessment ^1^	Posttreatment Assessment ^1^	F-Value	*p*-Value ^2^	η2
Food intake	3.3 ± 1.0	2.8 ± 1.1	8.64	0.040	0.313
Breathing	3.8 ± 0.6	3.5 ± 1	2.11	0.326	0.100
Speech	3.8 ± 0.4	3.2 ± 1	6.54	0.076	0.256
Pain	3.7 ± 0.8	2.3 ± 1.1	24.18	0.006	0.560
Mood	3.4 ± 0.8	3.1 ± 0.9	3.06	0.270	0.139
Mobility	3.8 ± 0.4	3.8 ± 0.4	0.00	>0.99	<0.001

^1^ Mean score ± standard deviation. ^2^ Holm–Bonferroni-corrected *p*-values.

**Table 6 cancers-13-05529-t006:** Frequencies of HNC-FIT verbal ratings.

	Score	Verbal Rating	Controls (*n* = 37)	Pretreatment (*n* = 24)	Posttreatment (*n* = 60)
Food intake	0	No oral feeding; only via gastrostomy tube	0	0	3
	1	Gastrostomy tube needed; some oral feeding possible	0	0	12
	2	No gastrostomy tube, oral diet, but only liquid/soft food	0	2	4
	3	No gastrostomy tube, diet/swallowing near-normal	2	7	15
	4	Normal	35	15	26
Breathing	0	Tracheostoma, needs blocked cannula	0	0	2
	1	Tracheostoma, speech cannula no cannula	0	1	8
	2	No tracheostoma, breathing difficult at rest	0	0	2
	3	No tracheostoma, breathing difficulties only on exertion	4	3	13
	4	Normal	33	20	35
Speech	0	Not possible/without phonation	0	0	4
	1	Difficult to understand, no phone calls	0	1	5
	2	Telephoning possible	1	0	5
	3	Easy to understand, but pronunciation/voice changed	3	6	16
	4	Normal	33	17	30
Pain	0	Pain despite opiate therapy	0	0	3
	1	Controlled with opiates	0	1	2
	2	Regularly needs non-opioid analgesics	1	2	5
	3	Needs analgesics from time to time	2	4	5
	4	Normal	34	17	45
Mood	0	Suicidal thoughts	0	0	0
	1	Very depressed despite antidepressants	1	1	3
	2	With antidepressants overall normal mood, very depressed without antidepressants	2	3	6
	3	Occasionally depressed, no antidepressants needed	4	5	15
	4	Normal	30	14	36
Mobility	0	Stiff neck and/or shoulder, hardly any movement possible	0	0	0
	1	Can hair hairdly comb, looking backwards in car not possible	0	0	3
	2	Combing with problems, looking backwards in car difficult	0	0	4
	3	Combing and looking backwards in car slightly restricted	5	2	13
	4	Normal	32	22	40

## Data Availability

The data presented in this study are available on request from the corresponding author.

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
