# Peer review of "A Tool for Rapid Assessment of Functional Outcomes in Patients with Head and Neck Cancer"

_cancers, 2021, doi:10.3390/cancers13215529_

Round 1
Reviewer 1 Report
In their study, Dejaco and colleagues developed a rapid physician-rated assessment of basic funcitonal outcomes in head and neck cancer patients. The study was carefully designed and appears to have been properly conducted. The work has relevance as I know from my own practice that questionnaires such as the EORTC-HN35-QoL are very comprehensive in their assessment. The use of a faster tool would be beneficial, so in principle I am positive about a publication.
However, I was very irritated by two things. First, the manuscript appears to be very sloppily prepared. Citations are missing everywhere or the tables are not really readable and therefore judgeable. Was this due to the authors or the editorial managment system of MDPI?
Second, I cannot understand at all why no oral and maxillofacial surgeon was involved in the review of the HNC-FIT scale. Since carcinomas of the oral cavity and food intake function are also assessed by the HNC-FIT scale, it MUST be mandatory that at least two oral and maxillofacial surgeons are also subsequently involved in an additional structured review (as part of the study). Otherwise, I do not see the manuscript as publishable.
Another limitation I see that it is not really clear when the data were collected temporally. It is important to standardize this. Especially using fixed time points (surgery, inpatient stay, radiation, chemotherapy, immunotherapy, etc.). I ask the authors to emphasize this clearly.
Furthermore, I recommend three things that are also necessary for the study to be publishable:
- It is mandatory to correct for multiple testing. There are too many P values.
- Performing double programming for statistics is necessary. That means, the core results have to be confirmed by a second software (R, SAS, Python or similar). Optimally if possible by a second person.
- I recommend to use a suitable reporting guideline from equator-network (https://www.equator-network.org/).
A limitation must also be built in. According to the manuscript, the questionnaires were in German. In Supplementary, both the German-language and the English-language translations must be presented. And something must be written into the manuscript as a limitation like "our questionnaire was tested on a German-speaking population. The English-language questionnaires have not yet been validated" or similar. Otherwise, there is a risk that the tool will be used worldwide in other languages without prior verification.
It must be clearly discussed what influence the stage, the type of therapy and the location of the tumor might have. And also point out the possible limitations there.
On formatting:
- The abbreviation AE occurs without defining it beforehand. This needs to be corrected in the introduction.
- Line 266: Citation error "Error! Reference source not found...". See also lines 277, 291, 297, 300, 332, 333, 391, 398, 403, 422, 426, 428, 431. Table 3 has some truncation "EORTC QoL ...?".
- Recommendation for Table 3: Order the EORTC QoL according to the cateogries where they most likely belong. Say, list everything that belongs to "Food intake" first, then what belongs to "Breathing", etc.
- There is also something wrong with the formatting in the references.
- You should capitalize ricc since it is an abbreviation.
- I cannot evaluate Supplemental data 3. Please correct the formatting and show it to me again.
- Aren't 120 sources a little too much for this type of study?
After making the appropriate changes, I would recommend "Accept". Until then, I recommend a "Major revision" so that the study improves significantly in quality.
Author Response
General
The authors very much appreciate the constructive and positive comments of the referee about the manuscript “A tool for rapid assessment of functional outcomes in patients with head and neck cancer” (ID cancers-1422330). All suggested corrections and changes are marked with track changes in the main manuscript. In addition, all supplemental data was moved to a separate file, as required by MDPI. As suggested additional semi structured interviews with two maxillofacial surgeons were performed and their valuable suggestions were included in the manuscript. Consequently, two additional authors have to be included in the present work. The authors would like to thank the reviewers for their time and effort since their suggested changes substantially improved the quality of the manuscript. Thank you very much for reconsidering our manuscript for publication in Cancers.
Referee 1
1.1.: “…the manuscript appears to be vey sloppily prepared. Citations are missing everywhere or the tables are not really readable and therefore judgable. Was this due to the authors or the editorial management system of MDPI?”
The authors very much apologies for the sloppy preparation of the manuscript. It appears that both citations and tables were not correctly converted from the raw manuscript to the template provided by MDPI. The entire manuscript was checked carefully and corrected accordingly, as suggested. Please also refer to response 1.10, 1.12, 1.14 and 1.15.
1.2.: “… I cannot understand at all why no oral and maxilofacial surgeon was invovled in the review of the HNC-FIT scale. Since carcinomas of the oral cavity and food intake function are also assessed by the HNC-FIT scale, it MUST be mandatory that at laest two oral and maxillofacial surgeons are also subsequently involved in an additional structured review (as part of the study).”
The authors very much appreciate this constructive suggestion. Consequently, two experienced maxillofacial surgeons were involved. Additional structured interviews were performed. The score for global acceptance was rated “1” and “2” by the two maxillofacial surgeons. Thus, the mean score for global acceptance (1.7) and the standard deviation (+/-0.6) did not change. However, both surgeons suggested adding chewing and dental status to the food intake domain. During the conceptualization, chewing and dental status were already considered relevant. Despite considerable effort, no meaningful operationalization (anchoring to external criteria, equidistance between verbal ratings) could be achieved. Consequently, chewing and dental status was subsumed under the higher-level functional domain of oral food intake. Both maxillofacial surgeons agreed on this subsumation and could be raised as co-authors. However, all authors involved are aware of this major limitation. Changes were made accordingly to the entire manuscript. Please refer to the Authors Section, line 5-6 and line 11, the Abstract, line 34, the Results section, line 243, 249-251, 252, 255-256, 258, and 264-266 as well as the Discussion section 380-381, 391, 514-523, 527-529. In addition, please refer to the Supplemental Data section, Supplemental Data 6 “Selection of HNC-related symptoms and functions not covered by HNC-FIT scales”, line 962-996.
1.3.: “… it is not really clear when the data were collected temporally. It is important to standardize this. Especially using fixed time points (surgery, inaptient stay, radiation, chemotherapy, immunotherapy, etc.) I ask the authors to emphasize this clearly.”
All authors apologize for this imprecision. Referee 2 also raised this issue. Consequently, the authors included the mean time intervals between the first and second assessment of the HNC-FIT-scale for pre-treatment and post-treatment patients. In addition, for post-treatment patients the interval since completion of treatment was provided. Finally, this limitation was added to the discussion section of the manuscript. Please refer to the Material and Methods section, line 195-197, the Results section, line 278-290 and the Discussion section, line 543-562. In addition, please also refer to response 2.1 of referee 2.
1.4.: “It is mandatory to correct for multiple testing. There are too many P values.”
We thank the reviewer for this important reminder. We applied the Holms-Bonferroni method for all p-values in case of multiple testing throughout the manuscript and added this information in the methods section. The applied correction of inflated p-values actually facilitated the interpretability (especially of the correlations with the EORTC H&N35 in table 3) and thus improved the readability of the manuscript. Please refer to the Materials and Methods section, line 221-222 and table 2, table 3, table 4 and table 5.
1.5.: “Performing double programming for statistics is necessary. That means, the core results have to be confirmed by a second software (R, SAS, Python or similar). Optimally if possible by a second person.”
The authors appreciate this constructive suggestion. We agree that in case of complex statistical analyses, double programming is a necessary step to avoid calculation errors. However, since in the present manuscript rather basic statistical standard procedures (correlation coefficients, analyses of variance) were applied which are default options in all mentioned statistical software, a double programming approach is highly unlikely to reveal different results. Due to limited time resources and to allow a timely response, we had not applied double programming in this case. We hope the editors and reviewer agree on this point.
1.6.: “I recommend to sue a suitable reporting guideline from equator-network (https://www.equator-network.org/).”
The authors appreciate this valuable suggestion. The homepage suggested was thoroughly searched for suitable reporting guidelines. However, to the best of our knowledge, currently no reporting guidelines for the development of an observer-rated tool for the assessment of functional outcomes in head and neck cancer patients are available. Instead, for the development and empirical validation of the HNC-FIT scales a step-wise approach was chosen as recommended by the Quality-of Life Group of the EORTC. Since this was previously not made clear in the manuscript, this was added accordingly. Please refer to the Materials and Methods section, line 91 to 94.
1.7.: “A limitation must also be build in. According to the manuscript, the questionnaires were in German. In Supplementary, both the German-language and the English-Language translations must be presented. And something must be written into the limitation.”
The authors very much appreciate this constructive suggestion. Referee 2 also raised this issue. The questionnaires were in German indeed since all patients and healthy participants involved in this study were German speaking. This was added as limitation to the discussion section accordingly. In addition, the original German-version of the HNC-FIT scales was provided as supplemental data. Please refer to the Discussion section of the manuscript, line 569-574 as well as the Supplemental Data section, Supplemental Data 8 “German Version of the adapted Head and Neck Cancer Functional Integrity Scale (HNC-FIT Scale), line 1001-1004. Please also see response 2.2. of referee 2.
1.8.: “It must be clearly discussed what influence the stage, the type of therapy and the location of the tuor might have. And also point out the possible limitations there.”
The authors very much agree on this issue. Stage, type of therapy and tumor site are very likely to influence the functional integrity scores raised via the HNC-FIT scale. However, the number of patients included in the study was too small to run multivariable analyses with the mentioned variables. The authors strongly agree that the influence of these confounders should be further explored in a larger sample of HNC-patients. As suggested, this limitation was added to the Discussion section of the manuscript. Please refer to the Discussion section of the manuscript, line 564-567.
1.9.: “On formatting: The abbreviation AE occurs without defining it beforehand. This needs do be corrected in the introduction.”
The authors apologize for this mistake. Since the abbreviation is only used once in the manuscript it was omitted. Please refer to the Introduction section, line 63.
1.10.: “On formatting: Line 266: citation error “Error! Reference source not found…”. See also lines 277, 291, 297, 300, 332, 333, 391, 398, 403, 422, 426, 428, 431.”
The authors are very sorry for all these formatting mistakes. It appears that both citations and tables were not correctly converted from the raw manuscript to the template provided by MDPI. All errors mentioned were checked and corrected accordingly. Please refer to the Results section, line 289-290, 302-303, 317-318, 326, 329, 340-341 and the Discussion section, line 359, 361-362, 420-421, 428, 432-433, 452-453, 456, 459 and 463. Please also refer to response 1.1.
1.11.: “… Table 3 has some truncation “EORTC QoL …?”. Recommendation for Table 3: Order the EORTC QoL accodring to the categories where they mos likely belong”
The authors apologize for the truncation of Table 3. It was reformatted accordingly and now reads ”EORTC QoL H&N35”. In addition, Table 3 was adapted as suggested, accordingly. Please refer to the Results section, Table 3, line 321 to 324.
1.12.: “There is also something wrong with the formatting of the references.”
The authors very much apologize for imprecision. It appears like Supplemental data 1: “Literature search of 120 included articles” was mistaken for the reference section of the manuscript. This list of studies yielded by the literature search performed was originally included in the MDPI draft of the manuscript, although it should have belonged in an additional file named “Supplementary data”. As required by MDPI, Supplemental data 1 was removed from the manuscript and all supplemental data is now provided in a separated file. To the best of our knowledge, no remaining formatting issues in the reference section could be identified. Please also refer to response 1.15 and the “Supplementary files” file, line 5-733.
1.13.: “You should capitalize ricc since it is an abbreviation".
The authors apologize for this mistake. “ricc” was capitalized to “rICC” as suggested throughout the manuscript. Please refer to the Discussion section, line 454, 455 and 459.
1.14.: “I cannot evaluate Supplemental data 3. Please correct the formatting and show it to me again.”
The authors very much apologize for the formatting mistake. Referee 2 also addressed this issue. In order to improve the readability of Supplemental data 3 “Adapted Head and Neck Cancer Functional Integrity Scale (HNC-FIT Scale)” it was changed from portrait-format to landscape-format. The authors hope that this improves the readability of Supplemental data 3. Please refer to Supplementary files, Supplemental data 3, line 741-750 as well as response 1.7.
1.15.: “Aren’t 120 sources a little too much for this type of study?”
The authors very much apologize for the confusion caused. As stated in response 1.12., it appears like Supplemental data 1: “Literature search of 120 included articles” was mistaken for the reference section of the manuscript. This list of studies yielded by the literature search performed was originally included in the MDPI draft of the manuscript, although it should have belonged in an additional file named “Supplementary data”. As required by MDPI, Supplemental data 1 was removed from the manuscript and all supplemental data is now provided in a separated file. Indeed, 120 sources would be too much for this type of study. As stated in the reference section of the manuscript, 30 original studies were cited in the entire manuscript. Since 4 additional sources were include, the manuscript now contains a total of 34 sources. Please also refer to response 1.12 and the References section of the manuscript, line 606-667.

Reviewer 2 Report
This proposal of a QOL scale in H&N cancers, which could be used routinely is very interesting, as the currently available tools are, as the authors state, too detailed and thus difficult to use in practice.
The methodology is appropriate, well described.
I have only a few comments:
- we have no indication on the moment when the questionnary was tested in the post treatment patients: answers may vary a lot according to the delay from the end of treatment. Could it be specified?
- This study was performed in german language. To be applicable in non german speaking countries, HNC-FIT scale should be translated and the translation should be validated. This should be specified in the article
- The authors made the choice to focus on most relevant domains, which they justify in the discussion, but a simple way to explore other aspects of QOL could have been, like in EQ5D scale, to add to their score a visual analogic scale between 0 and 100 to quantify a perceived health status. This could complete in a short and simple matter the range of this QOL evaluation. Could this be discussed?
- HNC FIT scale was impossible to read in the supplement 3, due to pagination problems. I could however understand it thanks to the instructions for clinicians in supplement 5
Author Response
General:
The authors very much appreciate the constructive and positive comments of the referees about the manuscript “A tool for rapid assessment of functional outcomes in patients with head and neck cancer” (ID cancers-1422330). All suggested corrections and changes are marked with track changes in the main manuscript. In addition, all supplemental data were moved to a separate file, as required by MDPI. As suggested additional semi structured interviews with two maxillofacial surgeons were performed and their valuable suggestions were included in the manuscript. Consequently, two additional authors have to be included in the present work. A “Change of Authorship Form” has been uploaded and signed by all authors as required by MDPI. The authors would like to thank the reviewers for their time and effort since their suggested changes substantially improved the quality of the manuscript. Thank you very much for reconsidering our manuscript for publication in Cancers.
Referee 2:
2.1.: “We have no indication the moment when the questionnary was tested in the post treatment patients: answers may vary a lot according to the delay from the end of treatment. Could it be specified?”
All authors very much agree on this issue. Referee 1 also raised this issue. Consequently, the authors included the mean time intervals between the first and second assessment of the HNC-FIT-scale for pre-treatment and post-treatment patients. In addition, for post-treatment patients the interval since completion of treatment was provided. However, most likely due to the small number of patients included in the study, no significant difference between post-treatment patients assessed early after treatment and late after treatment was observed. This limitation was added to the discussion section of the manuscript. Please refer to the Material and Methods section, line 195-197, the Results section, line 278-290 and the Discussion section, line 543-562. In addition, please also refer to response 1.4 of referee 1.
2.2.: “This study was performed in German language. To be applicable in non-German speaking countries, HNC-FIT scale should be translated and the translation should be validated. This should be specified in the article.”
The authors very much appreciate this constructive suggestion. Referee 1 also raised this issue. The questionnaires were in German indeed since all patients and healthy participants involved in this study were German speaking. This was added as limitation to the discussion section accordingly. In addition, the original German-version of the HNC-FIT scales was provided as supplemental data. Please refer to the Discussion section of the manuscript, line 569-574 as well as the Supplemental Data section, Supplemental Data 8 “German Version of the adapted Head and Neck Cancer Functional Integrity Scale (HNC-FIT Scale), line 1001-1004. Please also refer to response 1.7 of referee 1.
2.3.: “The authors made the choice to focus on most relevant domains, which they justify in the disussion, but asimple way to explore otheraspects of QOL could have been, like in EQ5D scale, to add to their score a visual analogic scale between 0 and 100 to quantify a perceived health status. This could be complete in a short an simple matter the range of this QOL evaluation. Could this be discussed?”
The authors very much appreciate this constructive suggestion. Adding additional domains including a quality of life domain was also suggested by some experts involved in the semi-structured interviews (Results section, line 258-261). In intense discussions, it was found that the suggested quality-of-life domain did not meet the main intention of capturing functional outcomes. Furthermore, it could not be operationalized by verbal ratings anchored to external criteria. This missing link to external criteria also applies to a visual analogic scale or the EQ5D. Thus, a quality- of-life domain was omitted in the HNC-FIT scales intentionally. Nevertheless, we strongly agree that this should be included in the discussion section and therefore was added, accordingly. Please refer to the Discussion section, line 514-521 and the Supplemental Data section, Supplemental Data 6 “Selection of HNC-related symptoms and functions not covered by HNC-FIT scales”, line 962-996.
2.4.: “HNC FIT scale was impossible to read in the supplement 3, due to pagination problems. I could however understand it thanks to the instructions for clinician in supplement 5.”
The authors very much apologize for the pagination problem with supplement 3. Referee 1 also addressed this issue. In order to improve the readability of Supplemental data 3 “Adapted Head and Neck Cancer Functional Integrity Scale (HNC-FIT Scale)” it was changed from portrait-format to landscape-format. The authors hope that this improves the readability of Supplemental data 3. Please refer to Supplementary files, Supplemental data 3, line 741-750 as well as response 1.7 and 1.14. of referee 1.

Round 2
Reviewer 1 Report
I thank the authors for their major revision and the interesting article and recommend accept in its current form.